# Effects of Transportation Stress on Complete Blood Count, Blood Chemistry, and Cytokine Gene Expression in Heifers

**DOI:** 10.3390/vetsci8100231

**Published:** 2021-10-13

**Authors:** Benito Avila-Jaime, Yareellys Ramos-Zayas, Moisés A. Franco-Molina, René Alvarado-Avila, Diana E. Zamora-Avila, Héctor Fimbres-Durazo, Juan J. Zárate-Ramos, Jorge R. Kawas

**Affiliations:** 1Universidad Autónoma de Nuevo León, Posgrado Conjunto Agronomiîa-Veterinaria, Avenida Francisco Villa S/N, Colonia Ex-Hacienda El Canadá, Escobedo, Nuevo León CP 66050, Mexico; benitoavilajaime@gmail.com (B.A.-J.); yareellysraza@hotmail.com (Y.R.-Z.); dna_100@hotmail.com (D.E.Z.-A.); hfimbres@outlook.com (H.F.-D.); jzarater@hotmail.com (J.J.Z.-R.); 2MNA de México, Avenida Acapulco 770, Colonia La Fe, San Nicolás de los Garza, Nuevo León CP 66477, Mexico; rene@mnademexico.com; 3Universidad Autónoma de Nuevo León, Facultad de Ciencias Biológicas, Laboratorio de Inmunología y Virología, Unidad C, Ave. Universidad S/N, Cd. Universitaria, San Nicolás de los Garza, Nuevo León CP 66455, Mexico; moyfranco@gmail.com

**Keywords:** transport stress, heifers, blood analysis, cortisol, cytokines, animal welfare

## Abstract

Blood samples were obtained from 16 high-risk heifers; eight were newly arrived from a 40 h road trip (0 days post-arrival (DPA)), whereas the other eight heifers had been in the feedlot at 25 DPA. Both groups were transported from the southeast tropical region of Mexico to a feedlot in the northeast and were sampled on the same day. The complete blood count, blood chemistry, and cytokine gene expression were analyzed. Gene expression was analyzed using specific primers to amplify and quantify the cDNA reverse transcribed from the mRNA transcripts for tumor necrosis factor (TNF)-α, interferon (IFN)-γ, and interleukin (IL)-2. Higher values for hematocrit (*p* = 0.029), hemoglobin (*p* = 0.002), eosinophils (0.029), albumin (*p* = 0.014), alanine aminotransferase (*p* = 0.004), bilirubin (*p* = 0.003), cholesterol (*p* = 0.014), and cortisol (*p* = 0.051) were observed in the 0 DPA group than the 25 DPA group. In the electrophoresis of TNF-α amplification products, two non-specific bands were observed in the 0 DPA group. These bands were sequenced, and BLAST analysis suggested that they corresponded to bovine lymphotoxin and have not been reported previously related to stress. The TNF-α expression level was higher (*p* = 0.001) in the 25 DPA group than the 0 DPA group according to the semi-quantitative expression analysis. This may indicate a persistent inflammatory process that could be related to trauma and disease, which can negatively impact their subsequent health and growth performance. In conclusion, homeostatic disruption was apparent in the 0 DPA heifers, which showed higher cortisol and reductions in TNF-α levels and stress-induced bovine lymphotoxin (SIBL) co-expression.

## 1. Introduction

Nearly all cattle are transported at least once in their lifetime [1]. When transported in poor conditions, long-distance transport can be considered one of the most stressful events for cattle due to the presence of adverse stimuli [2,3] that may negatively impact their health and growth performance [4]. Physical, psychological, and climatic factors affect cattle that are transported by road. Some aspects related to transport stress of cattle are previous nutrition and health, cattle overcrowding and mingling, poor handling, loading and unloading of the animals, irresponsible driving, drastic movements of the truck, bad road conditions, duration of the trip, lack of ventilation, water and food deprivation and extreme weather conditions [3].

Stress in cattle can alter physiological, hormonal, hematological, biochemical, and behavioral patterns, thus adversely affecting feed intake, reproduction, health, and production, as well as having effects at cellular and systemic levels [5]. Considering the immune response, transport stress results in increases in the number of white blood cells, neutrophils, eosinophils, and cytokine expression. Inflammatory cytokine signals are released during reactions to infection, tissue injury, and stress by immune cells, such as TNF-α, IFN-γ, and IL-2 [6]. This homeostatic disruption can immunosuppress the animals and make them more susceptible to disease [5].

Since the early days of farm animal industrialization, methods for reducing shrink and mortality have been sought to reduce the resulting economic losses. However, now we understand the importance of improving animal welfare for cattle productive performance and meat quality [7]. Transport stress can cause poor carcass quality, too low pH values, and possibly DFD meat when the animals are transported for too long or under inappropriate conditions [8]. Animal welfare has more recently become an important issue, since an increasing number of consumers are choosing products from farm animals that have suffered the least [9]. Producers are also inclined to reduce the suffering of animals, not only for ethical reasons but also to increase product quality, so that they can demand higher prices and reduce the economic losses and extra costs resulting from stress.

Understanding the effects of transportation stress under commercial conditions is necessary to define the most useful and effective management practices required for preconditioning cattle to reduce their suffering, improving their productive performance and quality of the meat that humans consume. To our knowledge, no study has reported the expression of cytokines in high-risk cattle, exposed to long-distance travelling. In this study, the high-risk cattle were undernourished; with no history of vaccinations or having been dewormed; were commingled with other lots immediately before been transported for an extended period of time without feed and water; and were highly stressed and prone to sickness. The heifers sampled in this study were either newly arrived at the feedlot from a 40 h road trip (0 DPA) or had recovered from shrinkage at 25 DPA after the same trip. The objective of this study was to determine the stress and health status (physiological and biochemical variables, and gene expression of TNF-α, IFN-γ, and IL-2) of high-risk heifers put through a long-distance road trip performed in poor welfare conditions at 0 DPA and 25 DPA.

## 2. Materials and Methods

This study was accepted by the Joint Graduate Program of the Faculties of Agronomy and Veterinary Medicine, at the University of Nuevo Leon.

### 2.1. Water Analysis and Quality

The total dissolved solids (TDS) in the water consumed by the receiving cattle were calculated by passing 50 mL of water through filter paper, which was then dried and weighed. The mineral concentrations in the water were determined using an Optima 2000 DV Optical Emission Spectrometer (Perkin Elmer Inc.; Waltham, MA, USA). The minerals analyzed in the water included Ca, Mg, Na, P, S, Fe, Mn, Zn, Cu, and Mo. The nitrate content was also determined (Standard Methods for the Examination of Water and Wastewater, 1995). The results from the physicochemical analysis of the water were as follows: TDS, 3,373 mg; Ca, 27 ppm; Na, 1,097 ppm; S, 360 ppm; Fe, 9.2 μg; Mn, 3.5 μg; Zn, 54.4 μg; Cu, 10.5 μg; and Mo, 41.6 μg. The nitrate concentration was low (6 ppm). The water contained high concentrations of sodium and sulfur.

### 2.2. Heifers, Transportation, and Reception Diet

Sixteen heifers from *Bos taurus* × *Bos indicus* crosses were randomly selected from two lots of cattle. The first group comprised 8 heifers that had just arrived from a 40 h road trip (0 DPA), with an average weight of 299 kg (±9.4 kg). The second group also comprised 8 heifers sampled on the same day as the first group, but they had arrived 25 days before (25 DPA), with an average weight of 296 (±11.1 kg). This group had already recovered from shrinkage.

The first group (0 DPA) had travelled 1600 km in a 40 h trip with an average outdoor temperature of 33 °C from a ranch in Chiapas, Mexico. The loading density was 0.90 m^2^/head. The truck was driven by a single operator the whole trip. At hour 28, the heifers were unloaded at a sanitary checkpoint, where water was offered, and then they were loaded again; this stop only lasted two hours. The second group (25 DPA) was from the same origin, and their whole trip was similar. The cattle were considered high-risk; they had access to water but were not offered feed before being transported. After arrival at the feedlot, all the heifers were unloaded and restrained in a hydraulic chute for routine weighing and physical check-ups. Both groups were sampled on the same day to prevent physiological alterations due to climate changes and by the same management personnel to exclude the possibility of technical inconsistencies.

The chemical composition of the heifers’ reception diet is presented in Table 1. The ration was offered twice daily at 8:00 and 16:00. The diet’s chemical composition was as follows: CP, 16.5%; NEm, 1.86 Mcal/kg; NEg, 1.21 Mcal/kg; NDF, 30.9%; ADF, 19.4%; ash, 5.3%; crude fat, 3.0%; and NFE, 44.3%.

### 2.3. Sampling Procedures

Blood samples were obtained from both groups on the same day to determine the complete blood count, blood chemistry, and cytokine gene expression. All the blood samples were realized via coccygeal venipuncture; the complete blood count samples were collected in Vacutainer^®^ EDTA tubes and the blood chemistry samples in Vacutainer^®^ tubes (Becton Dickinson and Company; Franklin Lakes, NJ, USA). The 0 DPA heifers were sampled in the morning on arrival, and the 25 DPA group was sampled before the morning meal. Samples intended for the analysis of gene expression were stabilized using an RNAlater^®^ solution (Life Technologies™; Carlsbad, CA, USA) according to the manufacturer’s protocol.

### 2.4. Complete Blood Count, Blood Chemistry, and Cortisol Assays

A complete blood count was conducted using a VetAutoread™ analyzer (Idexx Laboratories; Westbrook, ME, USA), which included measurements of hematocrit, hemoglobin, neutrophils, lymphocytes, the neutrophil–lymphocyte ratio (N:L), monocytes, eosinophils, neutrophils, and platelets. The plasma cortisol was determined by immunoassays performed using an IMMULITE^®^ 1000 analyzer (Siemens; Munich, Germany).

The plasma concentrations of glucose, blood urea nitrogen (BUN), creatinine, blood proteins (albumin, globulin, and total protein), bilirubin, and cholesterol, as well as the serum concentrations of the enzymes alanine aminotransferase (ALT), alkaline phosphatase (ALP), and amylase, were determined using the VetTest^®^ clinical blood chemistry analyzer (Idexx Laboratories; Westbrook, ME, USA).

### 2.5. RNA Extraction and Reverse Transcription—Polymerase Chain Reaction (RT-PCR)

RNA was extracted from blood using a RiboPureTM Blood Kit (Ambion^®^; Carlsbad, CA, USA) according to the manufacturer’s specifications and quantified using an EpochTM microplate spectrophotometer (Biotek^®^; Winooski, VT, USA). To observe RNA integrity, 50 ng of each RNA sample, based on spectrophotometric measurement, was analyzed by electrophoresis on a 0.8% agarose gel stained with GelRed (Biotium; Cat. No. 41003). The RNA concentrations of all the samples were standardized to 15.55 ng/µL. An RT-PCR was performed in a one-step reaction using the SuperScript^®^ III with a Platinum^®^ *Taq* DNA Polymerase kit (Invitrogen Co.; Carlsbad, CA, USA), following the manufacturer’s instructions, in a Maxygene™ thermal cycler (Axygen Scientific, Inc.; Union City, CA, USA). The primers used for TNF-α, IFN-γ, IL-2, and glyceraldehyde 3-phosphate dehydrogenase (GAPDH) (an internal control) amplification were previously described [6] and are presented in Table 2.

The amplified products were analyzed by electrophoresis on a 1.5% agarose gel and semi-quantitative values were assigned to the bands using the MYImage Analysis software (Thermo Fisher Scientific Inc.; Rockford, IL, USA).

### 2.6. Sequencing Non-Specific Bands

Non-specific bands were present in the 0 DPA group for TNF-α. After discounting all the possible errors that could have caused these bands, we extracted the DNA from the non-specific bands using the PureLink™ Quick Gel Extraction commercial kit (Life Technologies™; Carlsbad, CA, USA) following the manufacturer’s instructions. The DNA was then sequenced using an Ion Torrent™ sequencer (Life Technologies™; Carlsbad, CA, USA); the obtained sequences were analyzed using BLAST^®^ (National Library of Medicine; Bethesda, MD, USA).

### 2.7. Statistical Analysis

The hematological variables, blood biochemistry, serum cortisol, and cytokine gene expression were analyzed for normality using the Shapiro–Wilk test. Data were compared using either a two-sample t-test (non-rejection of the normality assumption) or a Wilcoxon rank-sum test (rejection of the normality assumption). A *p*-value < 0.05 was significant with respect to the normality test. For the paired t-test and Wilcoxon rank-sum test differences were considered significant at *p*-value < 0.05 using the Statistix^©^ 9 software (Tallahassee, FL, USA).

## 3. Results

### 3.1. Complete Blood Count

According to the complete blood count, hematocrit, hemoglobin, and eosinophils were lower in the 25 DPA group than the 0 DPA group (Table 3).

### 3.2. Blood Chemistry

The blood chemistry analysis showed higher concentrations of albumin, ALT, bilirubin, and cholesterol in the 0 DPA group than the 25 DPA group (Table 4).

### 3.3. Cortisol

In the 0 DPA group, the cortisol concentration was significantly higher than that in the 25 DPA group (Figure 1).

### 3.4. Cytokine Gene Expression

Upon TNF-α amplification by RT-PCR in the 0 DPA group, we found two non-specific bands in addition to the expected band (Figure 2). These bands were sequenced, and BLAST analysis suggested that they corresponded to bovine lymphotoxin with a 100% identity of the sequence compared to NCBI Z14137.1. This stress-induced bovine lymphotoxin (SIBL) has not been reported previously. The results of the quantitative expression analysis showed increased TNF-α expression in the 25 DPA group. The mean densitometry values for TNF-α were greater for the 25 DPA group than the 0 DPA group (Figure 3). The mean values for IFN-γ and IL-2 were not different.

## 4. Discussion

Understanding the effects of transportation stress under commercial conditions is important to establish cattle preconditioning and feedlot receiving management practices, improving their welfare and productive performance [11]. In this study, we found that a 40 h trip caused increased blood hematocrit, hemoglobin, eosinophils, albumin, ALT, bilirubin, and cholesterol levels in heifers of the 0 DPA group. The high concentration of albumin found in this group may suggest dehydration [12], which may be related to transport stress. Increased hematocrit levels can also indicate dehydration, which can be associated with water deprivation and water loss via respiration during transportation [13]. Heat stress due to a high ambient temperature during transportation leads to increased rates of respiration and water loss [14]. Parker et al. [15], however, did not find a change in hematocrit values in *Bos indicus* steers deprived of water for 90 h. This difference could have been the result of the climatic conditions and genetic variation in each experiment, as well as the transport factor, as these steers were not transported [15]. Cole and Hutcheson [16] also found increased blood hemoglobin in water-deprived and transported farm animals, which could have been the result of hemoconcentration due to dehydration. Difference in eosinophils values between groups could be induced by physical and emotional stress, which is thought to be attributed to elevated levels of plasma epinephrine and cortisol. Continuous release of corticosteroids or prolonged steroid therapy decreases eosinophil production [17].

Increased ALT values in blood might result from tissue damage, low perfusion in muscle tissue, reduced heat dissipation, hypoxia, and fatigue [18]. Bilirubin is commonly used as a biomarker of liver status in cattle and high levels are related with a more negative energy balance after long-distance transport, caused by a prolonged fasting period [19]. The high total cholesterol concentration before and just after transport might suggest that vitamin C decreased because of handling and transport stress. It is known that the metabolism of cholesterol needs vitamin C [20].

The increased serum cortisol reduced TNF-α and SIBL co-expression was obtained in heifers at 0 DPA. Cortisol is the most important hormone associated with stress, and in our study, the cortisol levels were higher in the 0 DPA group than in the 25 DPA group, suggesting high stress levels in the first group; most of the reviewed literature reports that cortisol increases in animals subjected to stressors [4]. In addition, some studies have shown that a mean value of > 70 ng/mL cortisol is possibly an indicator of rough handling [21], and a mean value over 60 ng/mL may reflect a level of fear or total panic in restrained animals [21]. A study on dairy cattle conducted to determine the effect of venipuncture in blood cortisol concentrations found a daily average range of 2.07–3.81 ng/mL in cows adapted to this handling practice; such levels can be considered the basal concentrations of this hormone [22]. Extreme cortisol levels of 93 ng/mL were obtained for cattle inverted on their backs [23].

Regarding the expression of cytokine genes, we found that TNF-α was significantly overexpressed in the 25 DPA group compared to the 0 DPA group. The TNF-α is involved in biological defense functions mediated by inflammation and is primarily produced by macrophages, lymphocytes, Kupffer cells, natural killer cells, and adipocytes. Some functions of TNF-α include antitumor and antimicrobial activity, mediation of inflammation, and regulating physiological functions such as appetite, fever, energy metabolism, and endocrine activity [24]. Viruses, parasites, other cytokines, and endotoxins induce TNF-α production. TNF-α is also related with the acute phase response (APR), activated by trauma, infection, stress, neoplasia, and inflammation. The APR is characterized by fever, leukocytosis, alterations in the plasma concentrations of trace minerals and hormones, changes in liver metabolism, and an increase in the plasma level of acute phase proteins [24]. The overexpression of TNF-α in the 25 DPA suggests that even after a recovery period from a long-distance journey, cattle are still undergoing an inflammation process that could be related to trauma or disease.

Though specific primers were used for TNF-α, SIBL expression was observed in the 0 DPA group. The literature indicates that both cytokines’ genes are in close linkage and tandemly arranged in the bovine genome, each with its own promoter and polyadenylation sequences [25]. Thus, under certain conditions, SIBL may be amplified by our TNF-α primers if it is expressed in the sample. In the case of the 0 DPA group of heifers with a higher level of stress, there may be factors that activate the expression of LT.

## 5. Conclusions

Homeostatic disruption and high stress levels were apparent in recently transported heifers exposed to poor welfare conditions observed during the trip from a cow-calf operation in southeast Mexico to a feedlot in the northeast. The transportation period for cattle should not exceed 18 h without rest and without offering water and feed, as established by the Official Mexican Norm for humane treatment in the mobilization of animals. Low-stress management and preconditioning practices might reduce the adverse effects of a trip conducted under the conditions of this study. Further studies should be conducted to determine the effect of transport in the SIBL expression, since it has not been reported in the transportation stress literature and may have potential as a biomarker.

## Figures and Tables

**Figure 1 vetsci-08-00231-f001:**
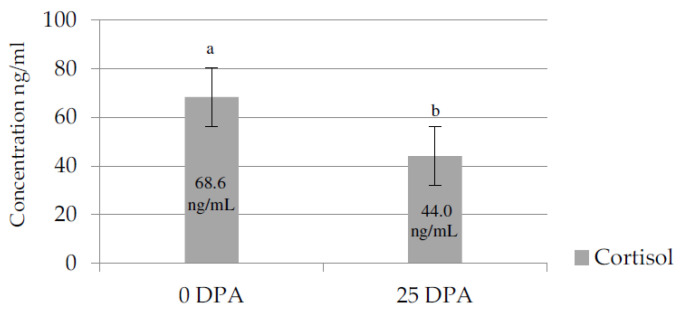
Cortisol serum levels (ng/mL) of heifers at 0 and 25 days post-arrival.

**Figure 2 vetsci-08-00231-f002:**
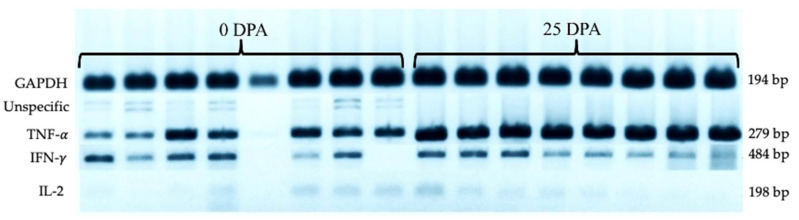
Agarose gel (1.5%) for TNF-α, IFN-γ, IL-2, and GAPDH. The first 8 columns correspond to the 0 DPA group, and the last 8 columns correspond to the 25 DPA group. Non-specific amplifications were only observed in the 0 DPA animals. The arrowhead indicates non-specific bands for TNF-α.

**Figure 3 vetsci-08-00231-f003:**
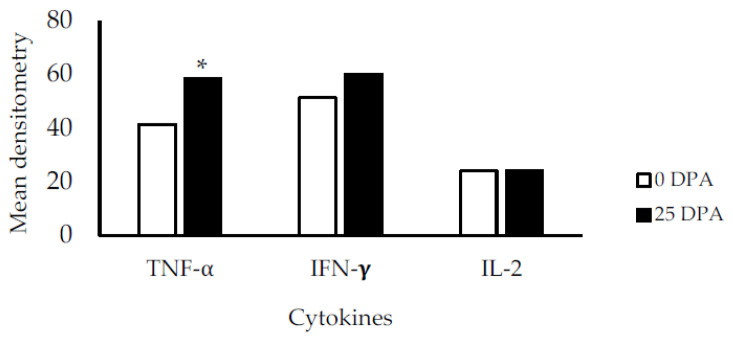
The mean densitometry values for TNF-α in the 0 and 25 DPA groups were 41.3 and 58.3, respectively (* *p* = 0.001). The mean values for IFN-γ were 51.3 for the 0 DPA group and 59.9 for the 25 DPA group (*p* = 0.461). The mean values for IL-2 in the 0 and 25 DPA groups were 24.0 in both cases (*p* = 0.990).

**Table 1 vetsci-08-00231-t001:** Heifers’ reception diet (as fed).

Ingredient	Content (g/kg)
Buffer grass hay	250
Steam-flaked sorghum	440
Distillers’ dried grain	100
Soybean meal	84.5
Whole cottonseed	50
Molasses	60
Mineral–vitamin premix ^1^	2.5
Calcium carbonate	8
Urea	5

^1^ Trace minerals (Mn, 17,000 mg/kg; Zn, 34,000 mg/kg; Cu, 3,400 mg/kg; I, 170 mg/kg; Co, 68 mg/kg; Se, 102 mg/kg); vitamin A (880,000 I.U./kg); vitamin E (20,000 I.U./kg); and sodium monensin (12 mg/kg).

**Table 2 vetsci-08-00231-t002:** Primer sequences used for amplification of the mRNAs for the cytokines TNF-α, IFN-γ, IL-2, and glyceraldehyde 3-phosphate dehydrogenase (GAPDH).

Gene	Sequence (5′ → 3′)	Amplicon (bp) ^1^	Annealing (°C)	Melting (°C)	Extension (°C)
TNF-α	F-TAACAAGCCGGTAGCCCACG	279	57	88.5	72
	R-TCTTGATGGCAGACAGGATG				
IFN-γ	F-GCCAAATTGTCTCCTTCTACTTC	484	61	85	72
	R-GGGTCAAGTGAAATAGTCACAGG				
IL-2	F-TCCAAGCAAAAACCTGAACC	198	57	82.5	72
GADPH	R-CAGCGTTTACTGTTGCATCATCF-GGCGTGAACCACGAGAAGTATAA	194	59	86	72
	R-CCCTCCACGATGCCAAAGT				

^1^ bp, base pairs.

**Table 3 vetsci-08-00231-t003:** Complete blood count values of heifers at 0 and 25 days post-arrival.

Analysis	Reference Values ^a^	Days after Arrival		SE	*p*
		0	25		
Hematocrit (%)	24–46	48.4	40.6	2.14	0.029
Hemoglobin (g/dL)	8.0–14.0	15.7	13.0	0.48	0.002
Neutrophils (%) ^b^	15–61	32.9	28.8	3.46	0.430
Lymphocytes (%)	26–68	54.0	61.0	3.91	0.227
Monocytes (%) ^b^	0–12	7.6	8.5	1.41	0.750
Eosinophils (%) ^b^	0–28	5.25	1.75	1.29	0.029
N:L ^b^	0.5	0.69	0.49	0.129	0.372
Platelets (K/µL)	175–500	572	688	125.1	0.521

^a^ Merck Veterinary Manual, 2010 [10]. ^b^ Shapiro–Wilk test (*p* < 0.05).

**Table 4 vetsci-08-00231-t004:** Blood chemistry values of heifers at 0 and 25 days post-arrival.

Analysis	Reference Values ^a^	Days Post-Arrival		SE	*p*
		0	25		
Glucose (mg/dL)	40–100	68.4	80.0	5.79	0.177
BUN (mg/dL)	10–25	12.0	9.25	1.059	0.088
Creatinine (mg/dL)	0.5–2.2	0.95	0.94	0.052	0.867
Phosphorus (mg/dL)	5.6–8.0	7.93	9.23	0.516	0.098
Calcium (mg/dL)	8.0–11.4	11.0	10.8	0.285	0.674
Total protein (g/dL)	6.7–7.5	7.29	7.26	0.207	0.934
Albumin (g/dL)	2.5–3.8	2.91	2.56	0.088	0.014
Globulin (g/dL)	3.0–3.5	4.63	4.71	0.108	0.579
ALT (U/L)	6.9–35	74.4	40.1	6.96	0.004
ALP (U/L) ^b^	18–153	153	136	26.15	0.958
Bilirubin (mg/dL) ^b^	0.0–1.6	1.1	0.49	0.104	0.001
Cholesterol (mg/dL)	62–193	100.3	66.6	8.46	0.014
Amylase (U/L) ^b^	41–98	56.8	12.5	8.41	0.139

^a^ Merck Veterinary Manual, 2010 [10]. ^b^ Shapiro–Wilk test (*p* < 0.05).

## Data Availability

The data presented in this study are available on request from the corresponding author.

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
