# Peer review of "Effects of Transportation Stress on Complete Blood Count, Blood Chemistry, and Cytokine Gene Expression in Heifers"

_vetsci, 2021, doi:10.3390/vetsci8100231_

Round 1

Reviewer 1 Report

No comment

Author Response

We appreciate your comments. We mades improvements to the paper.

Reviewer 2 Report

The manuscript "Effects of Transportation Stress in Complete Blood Count, Blood Chemistry, and Cytokine Gene Expression in Heifers" by Ávila et al. presents a study about a topic with importance for producers and consumers. It is a generally interesting approach. The introduction could be more on point. I think it is important to differentiate between transportation that takes place before the start of/in the middle of the fattening period and transport to the slaughterhouse. From what I understand, the present manuscript deals with the first scenario and this needs to be clearly specified in the introduction and discussion. The execution and presentation of gene expression data is insufficient and needs to be improved by following the MIQE guidelines. The presentation of the results should be improved. The conclusion is in part a summary of results and should be improved. Please see my more detailed comments below.

Graphical abstract:

Abbreviation "DPA" is unclear.

The arrow in front of TNFa is missing to indicate the direction of the difference.

L51: What does "high-risk" mean here?

LL57–59: The expression "concentrations of … hematocrit" does not work.

L62: Grammar: have instead of has.

LL63–66: Split in two sentences and correct grammar.

Key words: some of your key words occur already in the manuscript title. You might want to adjust that.

LL165–170: For how long after the transport finished are these increased numbers of white blood cells and increased cytokine expression observed? How fast do these normalize again? I understand that the issue is transport stress and how it affects the following productivity. However, almost all animals undergo some sort of transport before they are being slaughtered. Especially then e.g., the cytokine levels will be high. What you are referring to is transport that takes place during the fattening period. So it would be important to know for how long you can observe negative effects to see how much these actually impact on animal health/productivity/meat quality.

LL172–176: Please add information on how improved animal welfare affects performance and meat quality.

L185: Define abbreviation DPA at first occurrence in the main text.

LL279–283: Did you select 4 and 4 animals per lot per treatment group or 8 animals from lot 1 for one treatment and 8 animals from lot 2 for the second treatment? What was the breed of the animals?

Table 1: Please specify if the contents are given in fresh or dry matter.

L306: Later you state that you were using plasma samples for analyses. Please indicate which type of plasma (EDTA, heparin, …) was obtained.

LL307–308: How many hours were between samplings of the animals. Parameters such as cortisol show a diurnal rhythm, so if there was two much time between the samplings, this might have affected the results.

LL359–365: Did you analyse the RNA integrity (quality) before performing the RT-PCR? If so, how did you do it?

L366: Please refer to the MIQE guidelines on how to report gene expression results and what information to provide for this kind of analysis. I believe, GAPDH was not used as an "internal control" but as a reference gene. According to the MIQUE guidelines, one reference gene is not enough. Especially GAPDH has been shown to be regulated under different conditions and can thus not be regarded as a good choice for a reference gene.

LL444–446: Please add information how you tested for normal to distribution of the residuals to justify utilization of the t-test.

LL449 and the following: Do not repeat p-values from the table in the text. You can then simply state that it was significant difference.

Figure 2: Increase font size of "Figure 2" in the caption. I would recommend to insert a label in the figure that shows the reader without looking at the caption which samples belong to which experimental group. Please also remove the arrow and label of unspecific bands from the middle of the gel image and insert it on left side of the gel so that the reader can clearly see that there were really no unspecific bands in samples 9-16. Please also add figures (e.g., bar graphs like for cortisol, or even better: boxplots) for the densitometry results where you can indicate significant differences (and remove them from the figure caption).

LL597–708: Please insert information of diurnal cortisol rhythm and discuss how this can have (not) affected your results.

LL584 and following: I am missing discussion on many of your analysed parameters that showed significant differences, namely hemoglobin, amylase, ALT, bilirubin, and cholesterol. What do these changes mean? How do you explain them and what are the consequences? What have others reported in this context?

LL695–697: You did not investigate the effect of low-stress management and preconditioning practices in your study. Therefore, I would move this statement as an outlook towards the end and phrase it more like a possibility (could, might) and not like a given fact (can) because this needs to be investigated still, right?

L697: How do you conclude these 18 hours?

L689: You said that already in the first sentence of the conclusion.

LL699–703: This is more a summary of the results than a conclusion.

Author Response

We greatly appreciate your observations, we improved our article, and all the suggestions and comments were considered.

Answers to Reviewer 2:

Question (Q): Graphical abstract: Abbreviation "DPA" is unclear and the arrow in front of TNF-⍺ is missing to indicate the direction of the difference.

Answer (A): Graphical abstract observations have already been corrected.

Q: L51: What does "high-risk" mean here?

A: In this study, high-risk cattle were undernourished, with no history of vaccinations or been dewormed, commingled with other lots immediately before been transported for an extended period of time without feed and water, and highly stressed.

Q: LL57–59: The expression "concentrations of … hematocrit" does not work.

A: “Higher concentrations of … hematocrit”, was changed to “Higher values for … hematocrit”.

Q: L62: Grammar: have instead of has.

A: Changed “have” instead of “has”.

Q: LL63–66: Split in two sentences and correct grammar.

A: The TNF-α expression level was higher (p = 0.001) in the 25 DPA group than the 0 DPA group according to the semi-quantitative expression analysis. This may indicate a persistent inflammatory process that could be related to trauma and disease, which can negatively impact their subsequent health and growth performance.

Q: Key words: some of your key words occur already in the manuscript title. You might want to adjust that.

A: transport stress; heifers; blood analysis; cortisol; cytokines; animal welfare

Q: LL165–170: For how long after the transport finished are these increased numbers of white blood cells and increased cytokine expression observed? How fast do these normalize again? I understand that the issue is transport stress and how it affects the following productivity. However, almost all animals undergo some sort of transport before they are being slaughtered. Especially then e.g., the cytokine levels will be high. What you are referring to is transport that takes place during the fattening period. So it would be important to know for how long you can observe negative effects to see how much these actually impact on animal health/productivity/meat quality.

A: In group 0 DPA blood samples were obtain immediately post arrival. The 25 DPA treatment was considered to determine if 0 DPA values were normalized after 25 days.

     Yes, the issue is transport stress and its effects on physiological variables at 0 DPA and after 25 DPA. In our study, cattle were transported from a cow-calf operation located in Southeast Mexico to a feedlot in the Northeast. These animals were exposed to high-stress transportation conditions. Our interest was to measure these variables after the 25-days receiving period in the feedlot, before been offered the finishing diets. Generally, cattle recover their weight lost during transportation before 25 DPA. The purpose was to determine if normal physiological blood values were obtained after 25 DPA.

Q: LL172–176: Please add information on how improved animal welfare affects performance and meat quality.

A: We added “Transport stress can cause poor carcass quality, too low pH values and possibly DFD meat when the animals are transported for very long distance or under inappropriate conditions (Hartung et al., 2009)”.

Q: L185: Define abbreviation DPA at first occurrence in the main text.

A: It was defined in line 18 of the Abstract.

Q: LL279–283: Did you select 4 and 4 animals per lot per treatment group or 8 animals from lot 1 for one treatment and 8 animals from lot 2 for the second treatment? What was the breed of the animals?

A: We randomly selected 8 animals from one lot for 0 DPA and 8 animals from the second lot for the 25 DPA. The heifers were Creole (mixtures of Bos indicus cattle).

Q: Table 1: Please specify if the contents are given in fresh or dry matter.

A: The contents are given as fed.

Q: L306: Later you state that you were using plasma samples for analyses. Please indicate which type of plasma (EDTA, heparin, …) was obtained.

A: The blood was collected in EDTA tubes.

Q: LL307–308: How many hours were between samplings of the animals. Parameters such as cortisol show a diurnal rhythm, so if there was too much time between the samplings, this might have affected the results.

A: In 0 and 25 DPA groups blood samples were obtained in the morning, immediately post arrival, within minutes of each other.

Q: LL359–365: Did you analyze the RNA integrity (quality) before performing the RT-PCR? If so, how did you do it?

A: Yes, the integrity of the RNA was analyzed by spectrophotometry and by electrophoresis in agarose gel. The purity of the RNA was analyzed by the absorbance ratio of A260/280 obtaining a purity range in the samples of 1.8-2.0. Also, using spectrophotometry, the RNA concentration in each of the samples was determined.

Q: L366: Please refer to the MIQE guidelines on how to report gene expression results and what information to provide for this kind of analysis. I believe, GAPDH was not used as an "internal control" but as a reference gene. According to the MIQUE guidelines, one reference gene is not enough. Especially GAPDH has been shown to be regulated under different conditions and can thus not be regarded as a good choice for a reference gene.

A: The analysis of the expression of the study genes: TNF-⍺, IFN-?, IL-2 was made by RT-PCR in final point, not in Real-Time PCR, and GAPDH gene expression was analyzed as internal control gene of the PCR reaction, which at the same time, allowed us to evaluate the quality of the reaction in a single step.

Q: LL444–446: Please add information how you tested for normal to distribution of the residuals to justify utilization of the t-test.

A: After applying the Shapiro-Wilk test for normality, data were compared using either a two-sample t test (non-rejection of the normality assumption) or a Wilcoxon rank-sum-test (rejection of the normality assumption). A p-value < 0.05 was significant with respect to the normality test. For the paired t-test and Wilcoxon rank-sum- test a p-value of < 0.05 was significant.

Q: LL449 and the following: Do not repeat p-values from the table in the text. You can then simply state that it was significant difference.

A: The p-values from the text have been removed.

Q: Figure 2: Increase font size of "Figure 2" in the caption. I would recommend to insert a label in the figure that shows the reader without looking at the caption which samples belong to which experimental group. Please also remove the arrow and label of unspecific bands from the middle of the gel image and insert it on left side of the gel so that the reader can clearly see that there were really no unspecific bands in samples 9-16. Please also add figures (e.g., bar graphs like for cortisol, or even better: boxplots) for the densitometry results where you can indicate significant differences (and remove them from the figure caption).

A: The figure has been modified and the bar graphs has been added too.

194 bp

IL-2

198 bp

Figure 2. Agarose gel (1.5%) for TNF-α, IFN-γ, IL-2, and GAPDH. The first 8 columns correspond to the 0 DPA group, and the last 8 columns correspond to the 25 DPA group. Non-specific amplifications were only observed in the 0 DPA animals. The arrowhead indicates non-specific bands for TNF-α.

Figure 3. The mean densitometry values for TNF-α in the 0 and 25 DPA groups were 41.3 and 58.3, respectively (p = 0.001). The mean values for IFN-γ were 51.3 for the 0 DPA group and 59.9 for the 25 DPA group (p = 0.461). The mean values for IL-2 in the 0 and 25 DPA groups were 24.0 and 24.0, respectively (p = 0.990).

*

Cytokines

Q: LL597–708: Please insert information of diurnal cortisol rhythm and discuss how this can have (not) affected your results.

A: Results from cortisol analyzes were from a single sample obtained early in the morning from each heifer at 0 DPA or 25 DPA.

Q: LL584 and following: I am missing discussion on many of your analysed parameters that showed significant differences, namely hemoglobin, amylase, ALT, bilirubin, and cholesterol. What do these changes mean? How do you explain them and what are the consequences? What have others reported in this context?

A: Difference in eosinophils values between groups could be induced by physical and emotional stress, which is thought to be attributed to elevated levels of plasma epinephrine and cortisol. Continuous release of corticosteroids or prolonged steroid therapy decreases eosinophil production (Ishizaki, 2010).

Increased ALT values in blood might result from tissue damage, low perfusion in muscle tissue, reduced heat dissipation, hypoxia, and fatigue (Lopez et al., 2006). Bilirubin is commonly used as a biomarker of liver status in cattle and high levels are related with a more negative energy balance after long distance transport, caused by a prolonged fasting period (Marcato, 2021). The high total cholesterol concentration before and just after transport might suggest that vitamin C decreased because of handling and transport stress. It is known that the metabolism of cholesterol needs vitamin C (Ishiwata, 2008).

Q: LL695–697: You did not investigate the effect of low-stress management and preconditioning practices in your study. Therefore, I would move this statement as an outlook towards the end and phrase it more like a possibility (could, might) and not like a given fact (can) because this needs to be investigated still, right?

A: Thank you for the observation, it has been changed.

Q: L697: How do you conclude these 18 hours?

A: The Official Mexican Norm (NOM-051-ZOO-1995) for humane treatment in the mobilization of animals establishes that transportation period for cattle should not exceed 18 hours without rest and without giving them drinking water.

Q: L689: You said that already in the first sentence of the conclusion.

A: Thank you for the observation, it has been changed.

Q: LL699–703: This is more a summary of the results than a conclusion.

A: Thank you for the observation, it has been changed.

This manuscript is a resubmission of an earlier submission. The following is a list of the peer review reports and author responses from that submission.

Round 1

Reviewer 1 Report

The manuscript “Effects of Transportation Stress on Complete Blood Count, Blood Chemistry, and Cytokine Gene Expression in Heifers” is not suitable for publication on current form for the following main reasons – the introduction is brief and does not fully show the purpose of the study; the novelty of the study is not mentioned; the discussion does not fully coincide with the results of the study; no new knowledge was provided.

Reviewer 2 Report

General comments:

The manuscript “Effects of Transportation Stress on Complete Blood Count, Blood Chemistry, and Cytokine Gene Expression in Heifers” contains good scientific information. The paper is well articulated. However, there some  comments that authors must attend to

Comments

Introduction

Line 35, change the word “performed” to “transported”

Line 38-40: sentence “methods for reducing shrink 39 and mortality have been sought to reduce the” is unclear, revise

Table 2:  a new column with a well detailed PCR condition for each gene investigated should be included

Line 188: This sentence is clearly understood “a Merck Manual, Veterinary Manual, 2010”

Table 4: values heading seems to have mixed up and should be cross-checked.

Statistical analysis

It would have been helpful if the authors had compared the various variables like hemoglobin. WBC count, blood biochemistry etc, using Chi-square. It will boast the current analysis and further demarcate the difference that exists between 0 DPA group and 25 DPA group

Conclusions

The conclusion was well reported but lacked substantial recommendation. The authors remarked, “Low-stress management and preconditioning practices can reduce the adverse effects of a trip” They, however, did not give an example(s) of such Low-stress management and preconditioning practices.